# Local-Forest Method for Superspreaders Identification in Online Social Networks

**DOI:** 10.3390/e24091279

**Published:** 2022-09-11

**Authors:** Yajing Hao, Shaoting Tang, Longzhao Liu, Hongwei Zheng, Xin Wang, Zhiming Zheng

**Affiliations:** 1School of Mathematical Sciences, Beihang University, Beijing 100191, China; 2Key Laboratory of Mathematics, Informatics and Behavioral Semantics (LMIB), Beihang University, Beijing 100191, China; 3Institute of Artificial Intelligence, Beihang University, Beijing 100191, China; 4State Key Laboratory of Software Development Environment (NLSDE), Beihang University, Beijing 100191, China; 5Beijing Advanced Innovation Center for Future Blockchain and Privacy Computing, Beihang University, Beijing 100191, China; 6PengCheng Laboratory, Shenzhen 518055, China; 7Institute of Medical Artificial Intelligence, Binzhou Medical University, Yantai 264003, China; 8School of Mathematical Sciences, Dalian University of Technology, Dalian 116024, China; 9Beijing Academy of Blockchain and Edge Computing (BABEC), Beijing 100085, China

**Keywords:** complex network, online social network, spreading, influential nodes

## Abstract

Identifying the most influential spreaders in online social networks plays a prominent role in affecting information dissemination and public opinions. Researchers propose many effective identification methods, such as k-shell. However, these methods are usually validated by simulating propagation models, such as epidemic-like models, which rarely consider the Push-Republish mechanism with attenuation characteristic, the unique and widely-existing spreading mechanism in online social media. To address this issue, we first adopt the Push-Republish (PR) model as the underlying spreading process to check the performance of identification methods. Then, we find that the performance of classical identification methods significantly decreases in the PR model compared to epidemic-like models, especially when identifying the top 10% of superspreaders. Furthermore, inspired by the local tree-like structure caused by the PR model, we propose a new identification method, namely the Local-Forest (LF) method, and conduct extensive experiments in four real large networks to evaluate it. Results highlight that the Local-Forest method has the best performance in accurately identifying superspreaders compared with the classical methods.

## 1. Introduction

The development of the Internet and Web 2.0 has spawned many online social platforms, usually modeled as online social networks [1,2,3,4], which have become the main underlying platforms of information dissemination [5] and public opinion formation [6]. It is widely believed that some nodes in social networks have a stronger spreading influence, which is related to their privileged locations [4,7]. These nodes with outstanding spreading ability can affect and even dominate the spreading processes. These nodes are called superspreaders [8], influencers [4], influential spreaders [9], or influential nodes [10]. Finding influential nodes has an important impact on many issues related to spreading processes, such as innovation diffusion [11], viral marketing [12], decision making [13], predicting popular scientific publications [14], and so on. Therefore, identifying influential nodes is of practical significance and has attracted extensive attention from scholars.

Nowadays, there have been many methods for superspreader identification, which have become an important field in network science. The mainly used methods are centrality-based heuristic methods [4,15], namely centrality methods. Specifically, these methods use the ranking of nodes’ centrality measure values to predict their spreading influence ranking. Here, the spreading influence of a node is generally quantified by the final propagation range it causes as the source of propagation. The classical measures include degree centrality (the number of the neighbors of a node) [16], betweenness centrality (the number of the shortest path through a node) [17], closeness centrality (reciprocal of the average shortest path distance to a node) [18], PageRank (the probability of accessing each node when the random walk process on the network reaches the steady state) [19], ks index generated by the k-shell decomposition (here we call it k-shell centrality) [9], and so on. Due to the great success of the k-shell method in epidemic-like models, such as the Susceptible–Infectious–Recovered (SIR) model, there are many follow-up studies, including improvement based on information entropy [20,21], improvement based on gravity equation [22,23], the hybrid method [24], improvement based on neighborhood information [25,26], improved decomposition method [27,28], and so on [29,30]. Moreover, there are other centrality methods proposed continuously. The node propagation entropy measure [31] considers the neighbors within two hops and the clustering coefficients of nodes. The semi-global triangular centrality method [32] uses the triangular patterns around nodes to identify influential spreaders. The generalized gravity centrality [33] combines the clustering coefficient and degree to evaluate nodes based on the gravity formula. The coupling-sensitive centrality [34] identifies the superspreaders on multiplex networks. Centrality measures can not only be used to identify individual spreaders but also can be applied to identify multiple spreaders, which is an important problem in social networks and is also known as the Influence Maximization (IM) problem [1,12,15,35,36,37]. For example, centrality methods combined with some selection strategies can be directly used to identify multiple influencers [9,15,38,39]. Other methods for IM problems include greedy algorithms [35,40], community-based algorithms [37,41], optimization-based meta-heuristic algorithms [1,42], and so on [43,44].

Despite the great progress of the identification methods, we find that the effectiveness of them is usually validated under the assumption that the spreading dynamic follows the epidemic-like models. However, this dynamical assumption has some deviations from spreading processes in online social media. The main difference is the Push-Republish mechanism with attenuation characteristic, which widely exists when information diffuses [3,45,46,47,48]. In short, the Push-Republish mechanism [3,45] is that the online users receive the messages passively and then may republish the messages depending on how attractive the message is. In addition, Ruan et al. [47] think that online users decide whether to spread the information mainly according to its content. This inspires them to propose an information contagion model where individuals make the final decisions to spread or not when being exposed to the information for the first time. In other words, unlike the passive disease infection process, people have the initiative to keep their personal tendency in the information diffusion. Furthermore, Kempe et al. [46] propose the decreasing cascade model, which sets that people’s probability of spreading information decreases with repeated stimuli and reflects a ‘marketing-saturated’ phenomenon. Both the model of Ruan et al. and the model of Kempe et al. portray the special attenuation characteristic in online spreading processes. Note that the difference between dynamical models can significantly affect the effectiveness of identification methods [49]. For example, the previously well-concerned k-shell method, which performs well under the epidemic-like models, is not good enough in the rumor dynamics [50]. These naturally raise the following problems: What is the performance of classical identification methods under the above mechanisms? Are there new effective methods inspired by the characteristics of the propagations in online social media?

To solve these problems, we consider the Push-Republish mechanism and Ruan et al.’s model together to describe the online spreading processes. For the convenience of description, we name the process the Push-Republish (PR) model. We find that, compared to the case under the SIR model, the classical identification methods have a worse performance when identifying superspreaders under the PR model. Furthermore, inspired by the features of the PR model, we propose a new centrality measure called the Local-Forest (LF) measure. To verify its effectiveness, we apply the proposed LF method to four real online social networks and compare it with the classical methods. Results show that the LF method performs better.

The outline of the paper is as follows. Section 2 introduces the PR model and shows the essential difference between the PR model and the SIR model. Section 3 gives a detailed description of the proposed LF measure. In Section 4, extensive experiments are conducted to evaluate the performance of the LF method and some classic methods. Finally, Section 5 summarizes the main results of this paper and prospects for future research.

## 2. Push-Republish Model

### 2.1. Description of the Push-Republish Model

The information diffusion procedure in online social media can be described as the following steps [3]: A user releases a message. Then friends of the user will learn of this message under the notification of the online social site, and this mechanism is called “push”. Affected by the interesting degree of the message, some friends will comment, cite or reprint the message. This behavior is called “republish”. The above steps will repeat recursively for each user who chooses to republish the message. In addition, some dynamical models, such as the decreasing cascade model [46] and Ruan et al.’s model [47], consider the attenuation characteristic to describe the influence propagation in social networks. In the decreasing cascade model, the probability that a node becomes activated decreases as the times of being stimulated increase. It indicates that the first contact with information is the most promising to lead to dissemination. Ruan et al. [47] divide nodes into three states: susceptible, adopted, or immune. In their model, a susceptible node converts its state to the adopted state or immune state immediately after receiving the information and then keeps its state. In other words, each node can be activated only when it is exposed to the information the first time.

The Push-Republish mechanism can directly describe the propagation mechanism in online social media and is compatible with Ruan et al.’s model, which considers the attenuation characteristic. Thus, we take the Push-Republish mechanism and Ruan et al.’s model into consideration and redescribe the above dynamical processes. For convenience, we name it the Push-Republish model. Online media can naturally be represented by networks where nodes represent users’ accounts, and edges represent the relationship between them. We divide all nodes into two states: *S* (do not know the message) and *R* (know the message). Nodes in set $I_{t}$ represent those who publish the message at the time *t*, i.e., those who can spread the message. Set the initial time *t* as 0. Choose a single node *s* as the origin of the propagation, and obtain $I_{0}=\left\{s\right\}$. Set the propagation probability as β. The PR model is as follows:1.Step 1: At the time *t*, for node *i* in the set $I_{t}$, its neighbors in *S* state will become *R* state (i.e., the “push” mechanism). Meanwhile, such neighbors who receive the message for the first time will choose to republish the massage with probability β (i.e., the “republish” behavior). Add the neighbors who choose to republish the message to the set $I_{t+1}$.2.Step 2: Remove *i* from $I_{t}$.3.Step 3: Perform steps 1–2 until $I_{t}=\emptyset$, then perform step 4.4.Step 4: Update the propagation process to the next time step, i.e., set t=t+1. If there is no node in set $I_{t}$, the propagation process ends. Denote the ending time as the propagation duration *T*. Otherwise, repeat step 3.

If edges in the network are directed (suppose that the user *i* follows the user *j*, then there is an edge from node *i* to node *j* in the network), the neighbors mentioned in the above PR model refer to the in-neighbors of nodes. We record the number of nodes with the state *R* when propagation ends to represent the propagation range. It can be seen that nodes with the state *R* in the PR model are the combination of the adopted nodes and the immune nodes in Ruan et al.’s model. The PR model has two main characteristics. First, the PR model introduces the push mechanism, which is consistent with the message transmission mechanism in online social media. Second, if the node that did not republish the message meets the message again, it will keep its decision, i.e., different from the passivity of individuals towards virus infection, individuals have the initiative to choose whether to spread the information or not, reflecting the attenuation characteristic.

### 2.2. Comparison between PR Model and SIR Model

We further compare the PR model and the SIR model (see details in Appendix A) from two aspects: (1) How does information propagate under these two models? (2) How do the classical degree and k-shell methods (see details in Appendix B) perform under these two models? The network used in this section is the largest connected component of the Facebook network [51], which is an undirected network with 63,392 nodes.

We use the distance between the source node and the nodes that republish the message to compare the characteristic of the two specific propagation processes. Specifically, we select a fixed node chosen from the inner-most shell (i.e., nodes with the highest ks index) of the network as the propagation origin and use the PR model and the SIR model as the propagation model to conduct 10,000 simulations, respectively. To facilitate the comparison between the PR model and the SIR model, in the PR model, we say nodes in set $I_{t}$ are in state *I* at time *t* (thus state *I* coexists with state *R*). For each time *t*, we calculate the average distance and minimum distance from the nodes in the state *I* to the source node. Figure 1 presents the time evolution of the average distance and minimum distance of the PR model (see Figure 1a) and the SIR model (see Figure 1b). We find that in the PR model, the minimum distance is not less than 2 when t>1, while this index is equal to 1 for a long time in the SIR model. This phenomenon can be directly explained by the dynamic mechanisms. In the PR model, neighbors obtain the message from the source node, and then if they decide not to republish the message, they will keep their tendency. Thus, the edges between the neighbors do not participate in the propagation process, and the potential propagation paths in the neighborhood of the original node form a local tree-like structure. As a result, the minimum distance is not less than 2 when t>1. However, in the SIR model, messages can be delivered among neighbors of the original node, and the edges between neighbors lead to repetitive stimuli, which promote the propagation. In reality, the inner k-shell is a cluster of nodes that are closely connected with each other. Kitsak et al. [9] have pointed out that one of the reasons why the k-shell method performed well in the SIR model was that the neighborhood of nodes in the high k-shell enabled them to effectively sustain propagation in the early stage of the propagation process, and then the epidemic could fully develop. Both the analysis of Kitsak et al. [9] and our experimental results here indicate the positive effect of closely connected neighbors on the propagation processes of the SIR model and explain the superiority of the k-shell method. However, due to the connections between neighbors becoming redundant in the PR model, the recognition ability of the k-shell method in it might not be as great as the case in the SIR model. The next experiment will confirm this claim.

Next, we will compare the performance of the classic degree method and k-shell method in the PR model and the SIR model. By convention, we carry out 1000 times Monte Carlo propagation simulation experiments in the PR model and the SIR model for each node in the Facebook network to estimate the real spreading influence of nodes in the two models, respectively [9,52]. Specifically, we use the average number of the *R*-state nodes when the propagation ends over all corresponding simulations to represent the source node’s real spreading influence. Then, sort all nodes in descending order according to their spreading influence. In this paper, we use $RK_{PR}$ and $RK_{SIR}$ to represent the sorting of nodes under the PR model and under the SIR model, respectively.

First, for each node in the inner-most shell (i.e., nodes with the largest ks index), we compare its ranking of the spreading influence in the PR model and that in the SIR model, as shown in Figure 1c. The horizon axis means the ranking of the node in the ranking list $RK_{PR}$. The vertical axis represents the ranking gap, which is calculated by the ranking of the node in $RK_{SIR}$ minus the ranking of the node in $RK_{PR}$. It can be seen that rankings of these nodes in the SIR model are generally higher (i.e., possess smaller ranking numbers) than that in the PR model, which means that the k-shell method performs not so well in the PR model.

Furthermore, we use the imprecision function [9,20,52,53] to measure the difference in the spreading influence between the top [pN] nodes (*p*∈[0,1]) predicted by the recognition method and the top [pN] most influential nodes given by $RK_{PR}$ or $RK_{SIR}$. Here *p* denotes the ratio of nodes, and *N* denotes the number of nodes in the network. The imprecision function [9,52] ε(p) for a given recognition method is defined as the following Equation (Equation 1):(1)$$\varepsilon \left(p\right)=1-\frac{M(p)}{Meff(p)},$$
where M(p) is the average spreading influence of the top [pN] nodes in the ranking list $RK_{M}$ (represents the nodes’ rankling list identified by centrality methods), and Meff(p) is the average spreading influence of the top [pN] nodes in sorting $RK_{PR}$ or $RK_{SIR}$. For each value of *p*, the smaller the ε(p), the better the recognition effect of the method. Figure 1d gives the imprecision function of the degree method and the k-shell method under the PR model and the SIR model, respectively. It can be seen that the recognition ability of the k-shell method (solid curves) in the SIR model (green curves) is significantly better than that of the degree method (dashed curves). However, in the PR model (red curves), the k-shell method is obviously inferior to the degree method in the small *p* region. Furthermore, compared to the case in the SIR model, the performance of the k-shell method decreases in the PR model, especially when identifying the top 10%.

In summary, the unique Push-Republish mechanism with attenuation feature in online social networks delivers different evolutionary results from those caused by the classical epidemic-like models. In addition, the performance of the traditional effective k-shell method decreases in the PR model. Therefore, a more effective method is necessary.

## 3. Local-Forest Method

As the simulations in the above section show, in the PR model, messages will not propagate between neighbors of the source node. To some extent, this leads to a locally tree-like structure, which is consistent with the reality that the majority of cascades on online social media are small and are described by a handful of simple tree structures [2]. Specifically, in the PR model, each node only has one choice to decide whether to republish the message or not and keeps the decision. Such a characteristic indicates that edges between neighbors of the source node are redundant. Inspired by this thought, we propose the Local-Tree (LT) measure and the Local-Forest (LF) measure. The LT measure is an intermediate measure to calculate the LF measure, whose basic idea is to remove the edges in the neighborhood of nodes that are redundant for propagation. For undirected networks, the LT centrality of a node is the sum of the degree of its neighbors minus twice the number of edges between neighbors. The LF centrality of a node is the sum of the LT values of its neighbors. The LF centrality LF(i) of node *i* is calculated as:(2)$$LT\left(j\right)=\sum_{w\in N(j)}k\left(w\right)-2m\left(j\right),$$
(3)$$LF\left(i\right)=\sum_{j\in N(i)}LT\left(j\right),$$
where N(j) is the set of neighbors of node *j*, k(w) is the degree centrality of node *w*, and m(j) is the number of edges between neighbors of node *j*. For directed networks, only the in-neighbors of nodes are considered, and the in-degree of nodes is used to replace the degree. Moreover, subtract the number of edges itself but not double. This is because one edge in an undirected network is equivalent to two opposite edges in a directed network. The summation of neighbors’ LT measure in the LF measure highlights the importance of the node’s degree and makes use of more information. Calculating m(j) requires two levels of traversal of *j*’s neighbors, so the time complexity of calculating the LF centrality for all nodes in the network is $O(N{\langle k\rangle }^{2})$, where *N* is the number of the nodes in the network and 〈k〉 is the average degree of all of the nodes in the network. Average degree 〈k〉 of sparse networks is much smaller than N. Thus in large-scale sparse networks, the time complexity of the LF measure is similar to the degree.

We consider a small network [9] as an example to illustrate the calculation of the Local-Tree measure and Local-Forest method (see Figure 1e). We take node 3 in Figure 1e as an example to calculate the LT centrality and the LF centrality. First calculate $LT\left(3\right)=\sum_{w\in N(3)}k\left(w\right)-2m\left(3\right)=23-2\times 5=1$3, where $N\left(3\right)=\left\{1,2,4,5,6,11,14,15\right\}$. It can be found that when calculating the LT centrality of node 3, the information in the inner-most shadow area in Figure 1e is used. Subtracting the item 2m(3) is equivalent to deleting the dotted line edge. Then a local tree structure is formed in the first-order neighborhood of the node, which is why we call this measure Local-Tree centrality. Next, calculate $LF\left(3\right)=\sum_{j\in N(3)}LT\left(j\right)=LT\left(1\right)+LT\left(2\right)+LT\left(4\right)+LT\left(5\right)+LT\left(6\right)+LT\left(11\right)+LT\left(14\right)+LT\left(15\right)=8+8+19+14+13+13+8+8=91$. The red edges in Figure 1e contribute to the calculation of the LF centrality of node 3. It can be found that the LF measure contains the information of the node’s second-order neighbors.

Taking each node in the network shown in Figure 1e as the origin of propagation, we conduct one hundred thousand propagation simulations for each case based on the PR model. The first row in Table 1 arranges these nodes in descending order according to their real spreading influence and merges the nodes with equivalent topological positions in the network into the same column. Table 1 gives the degree centrality, k-shell centrality, LT centrality, and LF centrality of each node. The LF centralities of nodes in adjacent columns are significantly different, indicating that the LF measure can better discriminate the nodes. On the whole, the LF centralities of nodes decrease with the real spreading influence decreasing, which indicates the excellent performance of the LF method.

## 4. Simulation Results

### 4.1. Experimental Setup

We conduct experiments on four large-scale real online social networks to compare the LF method with five widely-concerned centrality methods, including the degree [16], k-shell [9], closeness centrality [18], PageRank [19], and Mixed Degree Decomposition (MDD) method [27]. We give a brief introduction to these methods in Appendix B. We investigate the friendship networks gathered from (1) Brightkite [54,55], (2) Facebook [51,54], (3) DouBan [54,56], and (4) Twitter [54,57]. Twitter is a directed network, and other networks are undirected networks. Each network used in this paper is obtained by extracting the largest connected component from the original data set and the largest weakly connected component for the directed network. To precisely estimate the real spreading influence of each node, we simulate the propagation process based on the PR model via independent Monte Carlo experiments. Take the average of the propagation ranges (i.e., the number of nodes in the state *R* when propagation ends) of the Monte Carlo experiments as the real spreading influence of the source node. Still use $RK_{PR}$ to represent the ranking (in descending order) of nodes based on experiments, and use $RK_{M}$ to represent the ranking list based on identification methods.

Table 2 lists some basic characteristics of the networks and the parameters in the experiments. *N* is the number of nodes in the network, *M* is the number of edges in the network, and 〈k〉 is the average degree of nodes in the network. MC is the number of Monte Carlo experiments for each node. Related works in recent years usually set MC as 100 [32,33,34,53] or 1000 [23,31,39,52]. A relatively small MC (i.e., 100) is acceptable for large networks (i.e., with N> 10,000) [32]. Here, we set larger MC parameters for each network as far as possible within the range of the computational power. β is the propagation probability in the PR model. Note that the position of the source node makes no difference in the final propagation size under a large value of β. Thus, complying with previous studies, we also select the value of β around its threshold. Considering that the PR model has no corresponding threshold, here we select a β slightly larger than the threshold of the SIR model ($\beta_{th}$). Here, the threshold of the SIR model is calculated as $\beta_{th}=\langle k\rangle /\langle k^{2}\rangle$ [26], where 〈k〉 is the average degree, and $\langle k^{2}\rangle$ is the second-order average degree. Under this parameter setting, the proportion of propagations with duration 1 (i.e., with T=1) in all propagation simulations in each network ($P_{T=1}$ in Table 2) is 78–92%. This is in line with the characteristic of propagation flow on social networks described by Goel et al. [2], i.e., across seven online domains they investigated, in which 73% to 95% of instances show no diffusion. Here the propagation duration is the time when the propagation ends. For example, when there no neighbor of the source node republishes the message, the propagation duration is recorded as 1.

### 4.2. Methods Evaluation

Based on the experiments introduced above, we use three indicators [8,9,27,52] to analyze and compare the recognition ability of each method, including the imprecision function, the recognition rate, and the Kendall coefficient.

#### 4.2.1. Imprecision Function

The definition of the imprecision function ε(p) is shown as Equation (Equation 1) in Section 2. For any value of *p*, the smaller the value of the imprecision function, the better the recognition effect of the method. Figure 2 presents the imprecision function of each method, with *p* ranging from 0.01 to 0.5, and each subplot corresponds to a real online network. We can observe that our LF method, represented by the red curve in all subplots, has the smallest value of the imprecision function, which indicates the great performance of our method. In particular, when identifying the top 10% of superspreaders, our LF method significantly improves the efficiency of identification compared to other classical methods. The closeness method has a good performance, which is near the LF method in the DouBan network and the Twitter network, but the time complexity of the closeness measure is higher [15,18]. In the other two networks, our LF method has a significant advantage in the efficiency of identification. We conducted the experiments on Ubuntu Linux 18.04 LTS equipped with Intel(R) Xeon(R) CPU E5-2690 v3 @ 2.60GHz, and the algorithms were implemented in Python 3.8.5. The average consumed time of the 10 runs for each method on the four large networks is shown in Table 3. The results show that the time complexity of the LF method is the second-smallest among these methods (only larger than that of the degree method). Considering both the performance and time complexity, our LF method is the best choice among these methods.

#### 4.2.2. Recognition Rate

The recognition rate reflects the ability of methods to find important nodes. Here we use Pei et al.’s [8] definition of the recognition rate r(p), which reads as:(4)$$r\left(p\right)=\frac{|I_{p}\cap P_{p}|}{|I_{p}|},$$
where $I_{p}$ is the set of nodes ranking in the top *p* fraction of the $RK_{PR}$, $P_{p}$ is the set of nodes ranking in the top *p* fraction of the $RK_{M}$, and $|I_{p}|$ is the number of nodes in $I_{p}$. For any value of *p*, the greater the recognition rate r(p), the higher the accuracy of the method for influential node recognition. Figure 3 shows the curves of the recognition rate of different methods in the four real networks, with *p* ranging from 0.01 to 0.5. It can be seen that the red curve corresponding to the LF method is higher than other curves, which indicates that the LF method has better recognition accuracy in terms of identifying specific nodes than other methods.

#### 4.2.3. Kendall Correlation Coefficient

Kendall’s tau correlation coefficient [58] is a statistic that measures the rank correlation of two random variables. We use it to quantify the rank correlation between the measure and the real spreading influence obtained from Monte Carlo experiments. There are three calculation forms of Kendall’s tau coefficient. Considering that the number of values of the spreading influence (i.e., an average of propagation ranges) is more than that of the centrality measures, we use $\tau_{c}$ [59,60] in our paper. For each node *i*, record its centrality $x_{i}$ and its real spreading influence $y_{i}$ as $\left(x_{i},y_{i}\right)$. For any pair of $\left(x_{i},y_{i}\right)$ and $\left(x_{j},y_{j}\right)$, they are said to be concordant if $sgn(\left(x_{i}-x_{j}\right)\times \left(y_{i}-y_{j}\right))=1$, discordant if $sgn(\left(x_{i}-x_{j}\right)\times \left(y_{i}-y_{j}\right))=-1$. If $sgn(\left(x_{i}-x_{j}\right)\times \left(y_{i}-y_{j}\right))=0$, they are neither concordant nor discordant. $\tau_{c}$ is defined as:(5)$$\tau_{c}=\frac{2(n_{c}-n_{d})}{n^{2}\frac{m-1}{m}},$$
where $n_{c}$ is the number of concordant pairs, $n_{d}$ is the number of discordant pairs, *r* and *c* are numbers of the values of the *x* and *y* respectively, and m=min(r,c). Clearly, the higher the $\tau_{c}$, the higher the ranking correlation between the two variables. Briefly speaking, the above imprecision function evaluates the method in terms of the real spreading range, the recognition rate evaluates methods from the aspect of the specific nodes, and here, the $\tau_{c}$ gives a concise quantification of the ranking correlation between the measures and the real spreading influence of the nodes from a global aspect. Table 4 gives Kendall’s $\tau_{c}$ correlation coefficient between each measure and the real spreading influence. Apparently, the LF measure we proposed has the highest $\tau_{c}$ in each network considered, which means that it has the highest correlation with the real spreading influence of nodes.

## 5. Discussion

Identifying superspreaders is of great significance in dealing with many real problems, especially related to the spreading processes [61,62], such as promoting innovation diffusion and controlling rumor spreading. Nowadays, a large number of identification methods have been proven to be effective on epidemic-like propagation models. However, these propagation models rarely consider the Push-Republish dynamics with attenuation characteristics, which exist widely in online spreading processes. This naturally leads to the question of whether classical identification methods perform well when considering the new dynamics. In addition, proposing new effective identification methods is always a meaningful and challenging problem.

To address these issues, we use the Push-Republish model to describe the message propagation process in online social media based on the work of Zhao et al. [3] and Ruan et al. [47]. In this model, the message is pushed to users passively, and users keep their spreading decision. Through model simulations, we find that a local tree-like spreading structure in the neighborhood of the source node occurs and that the vast majority of information diffusion will end in one step, which is consistent with the previous discovery [2]. These phenomena indicate that the edges between neighbors of the source node may be redundant, i.e., make no difference in diffusion. Then, we test the performance of some classical identification methods by simulating our PR model. Results show the poorer performance of the widely concerned k-shell method when identifying the top 10% of superspreaders. Considering the redundant edges and tree-like structure, we define the Local-Tree measure and propose the Local-Forest method to identify superspreaders in online social media. We apply our LF method and other classical identification methods to four real large networks. Through comparison, we highlight that the LF method performs the best in all networks from the aspect of all indicators, including imprecision function, recognition rate, and Kendall’s correlation.

The LF method is not limited to the PR model because it is proposed based on the consideration of the redundant edges and tree-like structure, which are common features in various domains such as spreading processes. It indicates that the LF method has the potential to accurately identify superspreaders in other domains with the above features. In the future, we will explore the extension of our method in these fields. Moreover, the combination of the LF measure and the research about multi-source node identification is worth studying.

## Figures and Tables

**Figure 1 entropy-24-01279-f001:**
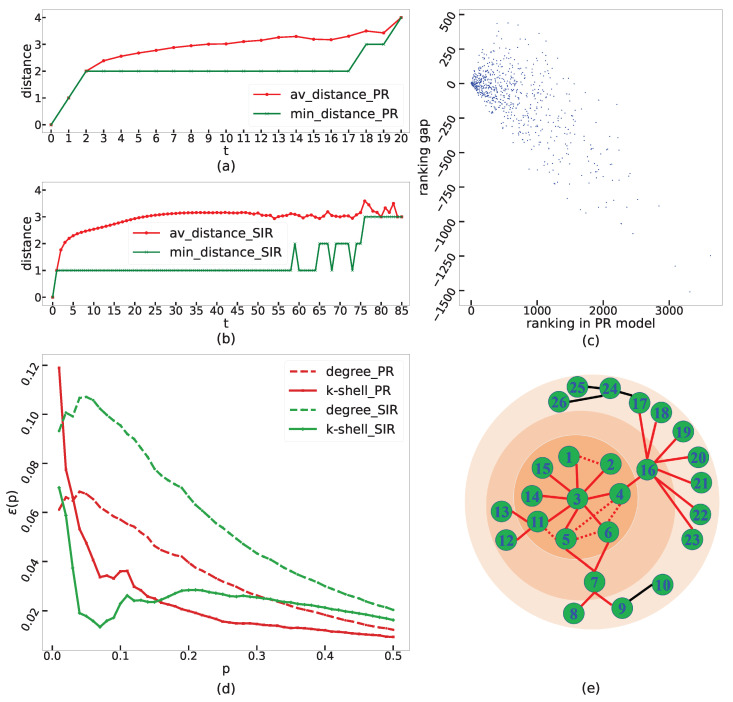
(**a**,**b**) The average distance and minimum distance from *I*-state nodes to the source node. (**c**) The ranking gap of nodes’ spreading influence between the case in the PR model and that in the SIR model. (**d**) The imprecision function of the degree method and the k-shell method under the PR model and the SIR model, respectively. We set β=0.012 in (**a**–**d**). (**e**) Schematic diagram of the Local-Tree measure and the Local-Forest measure.

**Figure 2 entropy-24-01279-f002:**
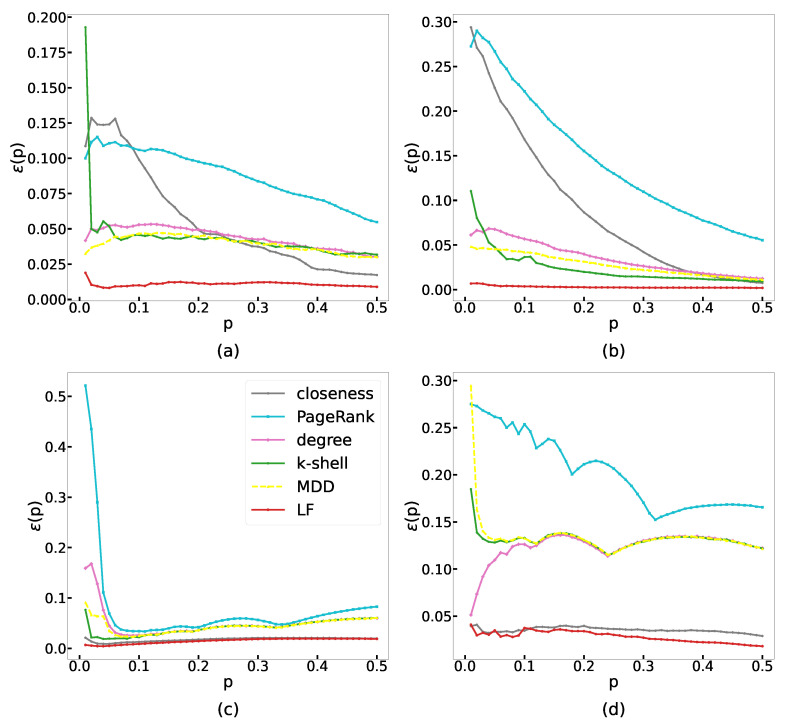
The imprecision function of the LF method and other classical methods under different *p* in four real networks, including (**a**) Brightkite, (**b**) Facebook, (**c**) DouBan, and (**d**) Twitter. Curves with different colors correspond to different methods: closeness (grey), PageRank (cyan), degree (pink), k-shell (green), MDD (yellow), and LF (red).

**Figure 3 entropy-24-01279-f003:**
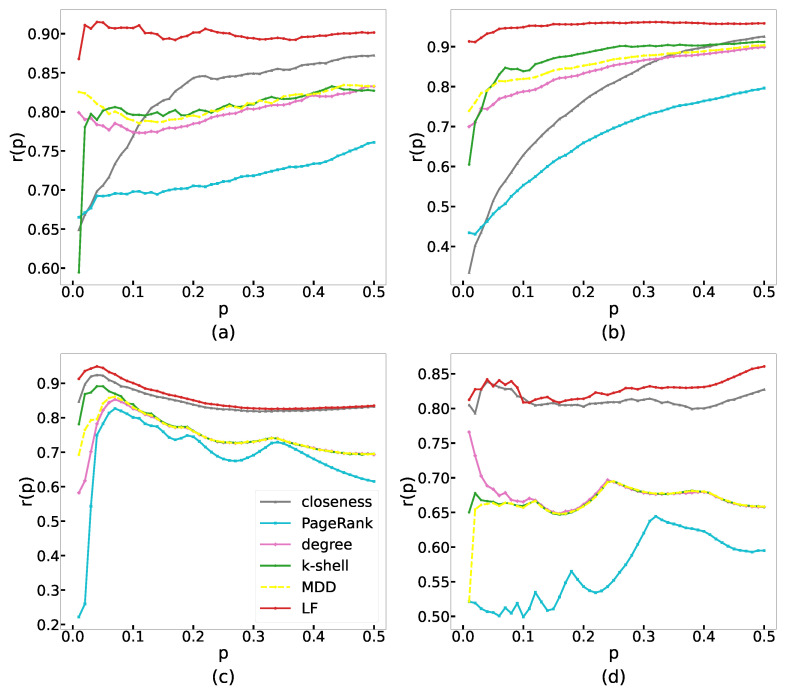
The recognition rate of the LF method and other classical methods under different *p* in four real networks, including (**a**) Brightkite, (**b**) Facebook, (**c**) DouBan, and (**d**) Twitter. Curves with different colors correspond to different methods: closeness (grey), PageRank (cyan), degree (pink), k-shell (green), MDD (yellow), and LF (red).

**Table 1 entropy-24-01279-t001:** The effectiveness of different identification methods in the sample of Figure 1e.

Centrality Measure	3	16	4	5	6	11	7	17	1,2	24	9	14,15	18–23	12,13	8	25,26	10
k-shell	3	1	3	3	3	2	2	1	2	1	1	1	1	1	1	1	1
degree	8	8	4	5	4	4	4	2	2	3	2	1	1	1	1	1	1
LT	13	12	19	14	13	13	10	11	8	4	5	8	8	4	4	3	2
LF	91	78	52	68	56	35	36	16	21	17	12	13	12	13	10	4	5

**Table 2 entropy-24-01279-t002:** Networks attributes, experimental settings, and some statistics for simulation results.

Network	N	M	〈k〉	MC	$\beta_{th}$	β	$P_{T=1}$
Brightkite	56,739	425,890	7.506	1000	0.015618	0.016	90.9%
Facebook	63,392	816,831	25.771	1000	0.011358	0.012	78.9%
DouBan	154,908	654,324	4.224	500	0.027100	0.028	91.8%
Twitter	465,017	834,797	1.795	200	0.116572	0.120	82.6%

**Table 3 entropy-24-01279-t003:** The consumed time of the LF method and other classical methods in the four real networks.

Network	Degree	k-Shell	Closeness	PageRank	MDD	LF
Brightkite	0.04799	1.73308	11,845.25111	14.45678	28.02683	0.95804
Facebook	0.03003	2.15947	24,302.80110	29.29709	44.02255	1.19346
DouBan	0.12007	1.50177	59,277.99047	27.76005	50.87384	1.72524
Twitter	0.24067	4.83157	1813.34151	14.19303	192.96852	2.10739

**Table 4 entropy-24-01279-t004:** $\tau_{c}$ between centrality measures and real spreading influence in four real networks. The best value is in bold (higher is better).

Network	Degree	k-Shell	Closeness	PageRank	MDD	LF
Brightkite	0.42768	0.45701	0.68762	0.22887	0.42658	**0.69770**
Facebook	0.79352	0.83248	0.80367	0.55818	0.80476	**0.90089**
DouBan	0.65134	0.65002	0.72248	0.45388	0.65896	**0.78665**
Twitter	0.34802	0.35625	0.63684	0.17299	0.34434	**0.70861**

## Data Availability

Not applicable.

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
