# Peer review of "Local-Forest Method for Superspreaders Identification in Online Social Networks"

_entropy, 2022, doi:10.3390/e24091279_

Round 1
Reviewer 1 Report
Identifying the influential spreaders in online social networks is an important (and challenging) problem. This paper proposed a new model, namely the Push-Republish (PR) model, to describe the spreading processes. Based on this, the authors proposed a new identification method, namely the Local-Forest (LF) method, and conducted extensive experiments in four real large networks to evaluate it. The paper is clearly written overall, however, there are some problems need to be addressed before publication. My comments are as follows:
1. For the Introduction part
In the Introduction, the authors rarely describe the related work in recent years, and most of the papers cited are classical methods. In addition, the authors also mentioned the problem of Influence Maximization, while only the centrality approach is mentioned.
2. For the Model part
In the Abstract, the authors claim that they propose a new information diffusion model, namely the Push-Republish (PR) model. However, I find that this model is very similar to the model studied by Ruan et al., see
Z. Ruan, J. Wang, Q. Xuan, C. Fu, G. Chen, Information filtering by smart nodes in random networks, Physical Review E, 98, 022308 (2018).
What is the difference between the PR model and the studied model?
Besides, there are also some problems in the description of the PR model. The meaning of the sentence is not clear. For example, “Remove the state I of node i.” This expression is difficult to understand.
3. For the Method part
This method only considers the local information of the target node. Many previous works have taken global information into account. Is that possible to consider the global information?
4. For the Experiment part
What do the parameters beta_c and TR_1 in Table 2 mean?
In page 8, the authors say “Considering both the performance and time complexity, our LF method is the best choice among these methods.” However, the authors did not conduct time complexity experiments.
Reviewer 2 Report
In the manuscript entitled "Local-Forest Method for Superspreaders Identification in Online Social Networks" is presented a novel measure, named Local-Forest (LF), to identify the most influential nodes in social networks. LF relies on the intuition that a news, originating at node i, is shared with the neighbouring nodes of i. This implies that in this respect connections between the neighbours of i are redundant and can be ignored. In this paper, LF is used in connection to a Push-Republish (PR) model of information diffusion in networks and compared to classical measures such as degree, closeness, PageRank, k-shell, mixed degree decomposition. The results show a clear advantage in using LF. The paper is plainly written and I enjoyed reading it. Nonetheless I suggest the following improvements prior to its acceptance for publication in Entropy: - At page 4 (line 137) it is claimed that 10000 simulations have been carried out with the PR and SIR (Susceptible-Infectious-Recovered). Some consideration on this number is in order - Analogously, at page 7 (Tabel 2) are reported the numbers of Monte Carlo simulations for each node of the test networks. Some considerations on these numbers are in order - In Figure 2c is shown a non monotonic behaviour of the imprecision function for the PageRank curve. In Figure 2d this behaviour is even more evident. I think that some consideration is necessary. Why, in the authors' opinion, such behaviour? - I am curious about the last paragraph of Section 5, as I wondered about how LF may perform in other models, for example the SIR model. Do the authors have some clue about it that could be anticipated in the closing remarks?Author Response
Please see the attachment.

Reviewer 3 Report
The authors presented a new metric PR to simulate the real importance of node. At the same time, they proposed a ranking method named LT and its improved version LF. Four real networks are used to evaluated the performance of the proposed method and to compare with other benchmark methods. Experimental results show that the proposed method outperform other classical methods. Generally, the topic of manuscript has some interesting, the manuscript is organized well. There are some problems should be considered.
(1) Why PR can describe the spreading in social media? It is better giving some refs or real examples.
(2) The description of PR is not clear. I think a double-loop process being used to simulate this spreading. The inner loop deal with the spreading from all I node at time t, and the outer loop is the iteration of time step.
Round 2
Reviewer 1 Report
The authors have successfully addressed my questions raised last time, I recommend it to be published after some further modifications in language.